# Enhancing Immunogenicity of Influenza Vaccine in the Elderly through Intradermal Vaccination: A Literature Analysis

**DOI:** 10.3390/v14112438

**Published:** 2022-11-03

**Authors:** Huy Quang Quach, Richard B. Kennedy

**Affiliations:** Mayo Clinic Vaccine Research Group, Division of General Internal Medicine, Mayo Clinic, Rochester, MN 55905, USA

**Keywords:** influenza vaccine, administration route, intradermal vaccine, intramuscular vaccine, subcutaneous vaccine, immunogenicity, older adults

## Abstract

Background: Aging and immunosenescence lead to a gradual decline in immune responses in the elderly and the immunogenicity of influenza vaccines in this age group is sub-optimal. Several approaches have been explored to enhance the immunogenicity of influenza vaccines in the elderly, including incorporating vaccine adjuvant, increasing antigen dosage, and changing the route of vaccine administration. Method: We systematically compared the immunogenicity and safety of influenza vaccines administered by intradermal (ID) route and either intramuscular (IM) or subcutaneous (SC) routes in older adults aged ≥ 65. Results: Of 17 studies included in this analysis, 3 studies compared the immunogenicity of ID vaccination to that of SC vaccination and 14 studies compared ID and IM vaccinations. ID vaccination was typically more immunogenic than both IM and SC routes at the same dosage. Importantly, a minimum of 3 µg of hemagglutinin antigen could be formulated in an ID influenza vaccine without a significant loss of immunogenicity. ID administration of standard-dose, unadjuvanted influenza vaccine was as immunogenic as IM injection of adjuvanted influenza vaccine. Waning of influenza-specific immunity was significant after 6 months, but there was no difference in waning immunity between vaccinations in ID, IM, or SC routes. While ID vaccination elicited local adverse reactions more frequently than other routes, these reactions were mild and lasted for no more than 3 days. Conclusions: We conclude that ID vaccination is superior to IM or SC routes and may be a suitable approach to compensate for the reduced immunogenicity observed in elderly adults. We also conclude that the main benefit of ID influenza vaccine lies in its dose-sparing effect. Additional research is still needed to further develop a more immunogenic ID influenza vaccine.

## 1. Introduction

Although the severity of influenza varies seasonally, older adults (aged ≥ 65) are at a higher risk of developing severe diseases and serious complications than younger groups [1]. Confounding this problem is the lower effectiveness of influenza vaccines in this age group [2]. As a result, up to 70% of hospitalizations and 85% of deaths associated with influenza occur in adults over 65 years of age [1]. This observation reflects the diminished immune responses in the elderly, an effect known as immunosenescence [3,4]. Altogether, these observations highlight an urgent need to develop a more immunogenic influenza vaccine for the elderly.

In order to improve the immunogenicity of influenza vaccine in the elderly, three main strategies have been investigated. The first strategy focuses on the incorporation of an effective vaccine adjuvant such as MF59 [5,6]. The second strategy involves increasing the amount of hemagglutinin antigen present in the vaccine [7]. The third strategy is to change the vaccine administration route from the conventional intramuscular (IM) or subcutaneous (SC) route to the intradermal (ID) route [8,9]. While MF59-adjuvanted and high-dose influenza vaccines are currently in use, ID influenza vaccine is not currently available for the elderly. Theoretically, ID vaccination takes advantage of the greater abundance of potent antigen-presenting cells in the dermis to achieve a similar level of immunogenicity at a lower dosage [9]. This dose-sparing effect is even more important during the pandemic where supply shortages may occur. While the immunogenicity of influenza vaccine at fractional dosages delivered by ID route and standard dose delivered by IM or SC routes have been reviewed elsewhere [10], the immunogenicity of influenza vaccine delivered by these routes in the elderly has not been systematically reviewed.

In this review, we aim to compare the immunogenicity and safety profile or reactogenicity of influenza vaccine in the elderly delivered by conventional IM or SC route and ID administration. To this end, this review aims to address following research questions: (i) Can ID administration induce superior immune responses at the same vaccine dosage delivered intramuscularly or subcutaneously in the elderly? In other words, can ID vaccination induce non-inferior immune responses at reduced dosages as compared to standard dose of influenza vaccine delivered intramuscularly or subcutaneously in the elderly? (ii) Is the immunogenicity of ID vaccination correlated to vaccine dosages? (iii) Is the immunogenicity of ID vaccination at standard dose comparable to those of high-dose or adjuvanted vaccines administered intramuscularly or subcutaneously? (iv) Is a two-dose regimen of influenza vaccination better than a one-dose regimen? (v) Are immune responses induced by ID vaccination as durable as those induced by IM or SC vaccination? (vi) Does ID administration elicit local and systematic adverse events more frequently and severely than IM or SC route?

## 2. Methods

### 2.1. Literature Search

We systematically searched all studies that used licensed vaccines and were published in English within the last 20 years using following keywords: influenza vaccine, intradermal, intramuscular, subcutaneous, intranasal, intravenous, microneedle, needle-free. We limited our review on (i) studies that were conducted under randomized clinical trials or included at least two groups of subjects receiving the same vaccine, but delivered via ID route and other routes; (ii) studies that used the same dosage of hemagglutinin antigen but administered by ID route and other routes (IM, SC…); (iii) ID vaccination at varying dosage of hemagglutinin antigen; (iv) studies that assessed the immunogenicity and reactogenicity or safety profile of influenza vaccines.

### 2.2. Data Extraction and Analysis

Raw data were extracted from all relevant studies, including the following 4 pieces of information: (1) the seroprotection rate (SPR), (2) the seroconversion rate (SCR), (3) the geometric mean titer (GMT), and (4) the frequency of both local and systemic adverse events. SCR is defined as percentage of vaccine recipients with fourfold increase (above 1:40) in influenza-specific antibody titer after vaccination as quantified by hemagglutination inhibition (HAI) assay, while SPR is defined as percentage of vaccine recipients with HAI titer ≥ 1:40 post vaccination. In studies examining the immunogenicity of influenza vaccines in different age groups, only data from older adults (aged ≥ 65) were extracted. For studies examining different vaccination routes, the ratios of SPR, SCR, GMT, and frequency of adverse events were calculated by dividing the rates elicited by ID vaccination over the rates induced by the other administration route(s) in the same study. One sample *t*-test (against a hypothesized ratio of 1) was applied to examine statistical significance of these ratios. A ratio of >1 means the immunogenicity or the frequency of adverse events higher by ID route than by IM or SC routes. All figures were plotted by GraphPad Prism version 9.1.0 (221) from GraphPad Software (San Diego, CA, USA).

## 3. Results

### 3.1. Characteristics of the Included Studies

An initial literature search resulted in 589,828 articles containing the above-mentioned keywords. Of these articles, 44,153 articles were conducted under randomized controlled trial or contained at least two groups of participants receiving the same vaccine via different administration routes. After screening titles and abstracts, 86 articles met one of the four above-mentioned requirements and underwent a full text assessment. This assessment resulted in 17 studies that were included in this analysis. Of the included studies, 3 studies compared the immunogenicity and reactogenicity of influenza vaccines administered by ID and SC routes [11,12,13]; 14 studies compared the immunogenicity and reactogenicity of ID and IM vaccinations [8,14,15,16,17,18,19,20,21,22,23,24,25,26]. Two studies evaluated the immunogenicity of one dose versus two doses of influenza vaccines administered intradermally and subcutaneously [12,13]. Five studies evaluated the duration of immune response at three or six months after vaccination [13,14,17,21,23]. Six studies compared the immunogenicity of ID influenza vaccine at standard dose and adjuvanted, standard-dose influenza vaccines injected intramuscularly [14,21,22,23,24,26]. Five studies evaluated the immunogenicity of ID influenza vaccine at varying dosages [12,18,20,21,24].

Our review focused on the participating subjects that were ≥65 years old (Table 1). Although the age of study subjects was slightly different among studies, subjects in vaccine groups of each study had similar average/median ages and age distributions (Table 1), minimizing a potential confounding effect of age on the vaccine-induced immunogenicity as observed earlier [27]. These studies included both female and male participants, although a potential confounding effect of gender on immunogenicity was not examined in these studies. Other demographic and clinical characteristics of study subjects, such as race, ethnicity, and health status, were also not considered as covariates in these published studies.

### 3.2. Is ID Vaccination More Immunogenic than SC or IM Route?

Of the included studies, three studies compared the immunogenicity of ID versus SC vaccination [11,12,13]; eleven studies compared the immunogenicity of influenza vaccines administered via ID and IM routes [14,15,16,17,18,19,21,22,23,24,25]. All of the included studies used trivalent influenza vaccine, containing a standard dose of 15 µg of hemagglutinin antigen from each of the H1N1, H3N2, and B vaccine strains. The immunogenicity was evaluated at 21–30 days post vaccination (Table 1) and expressed as seroprotection rate (SPR), seroconversion rate (SCR), and geometric mean titer (GMT). We calculated the ratios of SPR, SCR, and GMT elicited by ID vaccination over IM or SC routes, with a ratio of >1 meaning the immunogenicity was higher by the ID route than by IM or SC route.

In each of the included studies, the immunogenicity induced by ID vaccination was significantly higher than that by IM or SC vaccination [11,12,13,14,15,16,17,18,19,21,22,23,24,25]. In terms of the immunogenicity ratio, ID vaccination was still significantly more immunogenic than IM or SC vaccination, despite a wide variation in the immunogenicity ratios among these studies (Figure 1). As such, GMT ratios of >1.5 were found in ID over SC vaccination, suggesting a significant increase of at least 50% in GMT by switching from the SC to ID administration route (Figure 1A). Similarly, GMT ratios were also significantly higher by the ID route than by IM route, with the ratios of 1.26, 1.28, and 1.27 for H1N1, H3N2, and B strains, respectively (Figure 1B).

Meanwhile, the SPR ratios of 1.08, 1.05, and 1.56 were calculated for H1N1, H3N2, and B strains, respectively, all of which were statistically significant (Figure 1A). The SCR ratios also varied for each vaccine strain, with SCR ratios of 1.25 and 1.49 for H1N1 and B strains, respectively, being significantly higher by ID than SC route. The SCR for H3N2 strain varied largely at an average of 1.26 but was not significantly higher in ID vaccination than in the SC route (Figure 1A). In contrast, SCR ratios of ID over IM vaccination were 1.33, 1.41, and 1.31 for H1N1, H3N2, and B strains, respectively, all of which were statistically significant (Figure 1B). Lower average SPR ratios of 1.09, 1.08, and 1.07 were found for H1N1, H3N2, and B strains, respectively, with the SPR ratios of ID over IM vaccination for H3N2 and B strains being not statistically significant (Figure 1B).

### 3.3. Is the Immunogenicity of ID Influenza Vaccine Dependent on Its Antigen Dosage?

Dose-sparing is one of the main advantages of ID vaccination. As shown in a previous study [9], ID influenza vaccine at reduced antigen dosages was as immunogenic as IM vaccine with a standard dosage of 15 µg hemagglutinin antigen per strain. Having confirmed that ID vaccination is more immunogenic than SC or IM vaccination at the same antigen dosage (Figure 1), our next question is whether the immunogenicity of ID vaccination is dependent on the amount of hemagglutinin antigen. Eventually, we aim to determine a minimal amount of hemagglutinin antigen needed in ID influenza vaccines that can induce non-inferior immune responses as compared to the standard dose of 15 µg hemagglutinin antigen.

Of the included studies, three studies compared the immunogenicity of ID vaccination at reduced antigen dosages of 6 µg (ID6) [11], 7.5 µg (ID7.5) [21], and 9 µg (ID9) [12] with standard-dose ID influenza vaccine (ID15). As shown in Figure 2A, ID influenza vaccines at reduced antigen dosages (ID6, ID7.5, ID9) induced non-inferior influenza-specific immune responses to ID15 [11,12,21]. Although SCR and GMT ratios were <1 for almost all viral vaccine strains (H1N1, H3N2, and B), SPR ratios were ≥0.94, suggesting that the amount of hemagglutinin antigen could be reduced to 6 µg without a significant loss of immunogenicity (Figure 2A). There were no significant differences in immunogenicity induced by ID6, ID7.5, and ID9 (Figure 2A). Interestingly, one study comparing the immunogenicity of ID influenza vaccines formulated with 3 µg (ID3) or 9 µg (ID9) of hemagglutinin antigen found that the GMT and SCR elicited by ID3 and ID9 were comparable and non-inferior to the standard dose of 15 µg hemagglutinin antigen given IM (IM15) [20]. Taken together, results from these studies imply that a minimal amount of 3 µg of hemagglutinin antigen could be formulated in an ID influenza vaccine, which can induce non-inferior immunogenicity compared to IM15. This would increase the number of available vaccine doses five-fold, while maintaining the necessary levels of immunogenicity.

A higher antigen dosage present in ID influenza vaccine was also explored for a further increase in immunogenicity [18,24]. Although ID influenza vaccine formulated with 21 µg of hemagglutinin antigen (ID21) appeared to be more immunogenic than ID15, the difference was not significantly different with the highest SPR, SCR, and GMT ratios of 1.08 (Figure 2A).

### 3.4. Is the Immunogenicity of Influenza Vaccines Delivered by ID Route Non-Inferior to Adjuvanted or High-Dose Influenza Vaccines Administered Intramuscularly?

Influenza vaccines containing either the adjuvant MF59 [5,6] or a higher dosage of hemagglutinin antigen (60 µg of hemagglutinin antigen per strain) [7] were formulated to enhance the immunogenicity of influenza vaccine in the elderly. Multiple publications have demonstrated that these vaccines also have improved efficacy/effectiveness [28,29,30,31], making them better comparators for assessing ID vaccination in this age group. Our next question is whether ID vaccination at standard dose induces non-inferior immunogenicity to IM administration of adjuvanted or high-dose influenza vaccines. Five studies compared the immunogenicity of ID15 and MF59-adjuvanted IM15 [14,21,22,23,26]; two studies compared the immunogenicity of ID15 and high-dose IM vaccine either with 45 µg (IM45) or 60 µg (IM60) of hemagglutinin antigen per strain [21,24]. In the studies comparing the immunogenicity of ID15 and MF59-adjuvanted IM15, the immunogenicity of ID15 and adjuvanted IM15 was comparable with the SPR, SCR, and GMT ratios slightly varying around 1 for all three viral vaccine strains (Figure 2B). These results suggest that a level of immunogenicity elicited by MF59-adjuvanted IM15 could be achieved by ID15; in essence, changing the administration route from IM to ID is as effective as adding MF59.

Meanwhile, ID15 was significantly more immunogenic than IM45, with the SPR, SCR, and GMT ratios of ≥1 for all three vaccine strains (Figure 2B) [21]. However, when compared to IM60, the immunogenicity of ID15 was significantly inferior, with the SPR, SCR, and GMT ratios varying from 0.98 to 0.58 for three viral vaccine strains (Figure 2B) [24].

### 3.5. Does Two-Dose Regimen Induce Stronger Immune Responses than One-Dose Regimen?

While a two-dose regimen of vaccination is routinely administered for various vaccines, a single dose of influenza vaccine is used seasonally. In fact, the Centers for Disease Control and Prevention (CDC) only recommends two doses of influenza vaccine for children aged 6 months to 8 years who have not previously received seasonal influenza vaccine [32]. We next examined whether a two-dose regimen of influenza vaccination is more immunogenic than a one-dose regimen in the elderly.

Of the included studies, two compared the immunogenicity of ID and SC vaccination after one dose and two doses of influenza vaccines in the elderly [12,13]. As shown in Figure 3A, the two-dose regimen did not enhance the immunogenicity of influenza vaccines over the one-dose regimen either by the ID or SC administration route (Figure 3A). In fact, the SPR and GMT after two doses were even lower, but not significantly, than those of after one dose, with the SPR ratio and GMT ratio < 1 (Figure 3A). The SCR ratios largely remained < 1 (Figure 3A), confirming a non-superior immunogenicity of two-dose regimen as compared to one-dose regimen.

Interestingly, the immune responses induced by a two-dose regimen appeared to wane slower than those by a one-dose regimen of ID vaccination, where GMT ratios for H3N2 and B were 1.38 and 1.51 at 90 days (Figure 3B); and 1.31 and 1.54 at 180 days after the last immunization (Figure 3C). However, this trend was not observed with H1N1 strain for either the ID or the SC route (Figure 3A,B). A difference in pre-existing antibodies against each viral strain potentially contributes to this differential waning.

### 3.6. Are Immune Response Induced by the ID Route Durable?

Having shown that ID vaccination is more immunogenic than SC or IM vaccinations at the same antigen dosage, the next question is whether the immune responses induced by ID vaccination are as durable as those induced by SC or IM vaccination. Five studies evaluated the duration of influenza vaccines administered intradermally and intramuscularly [13,14,17,21,23]. As shown in Figure 4A, the immune responses significantly waned after 3 months with the titers of influenza-specific antibodies being approximately 80% of their peaks at day 21–28 after vaccination. At three months post-vaccination, there was no significant difference in SPR, SCR, and GMT ratios between ID and IM vaccinations, suggesting that the influenza-specific immune responses induced by ID vaccination are as durable as those induced by IM vaccination (Figure 4A). When data for a longer timeframe (6 months post-vaccination) were evaluated, the duration of immune responses induced by ID and IM vaccination remained comparable (Figure 4B). One study has found that the immune responses 6 months after vaccination are comparable to pre-vaccinated titers, further highlighting the waning of immune responses against influenza [23]. None of the included studies evaluated the immunogenicity of influenza vaccine more than 6 months after vaccination. This result suggests that ID vaccination may not provide any increased level of protection against disease compared to vaccination by the IM or SC route. We also noted that among the three viral strains in the vaccine, the H1N1-specific immune responses appeared to wane faster than H3N2- and B-specific ones (Figure 4).

### 3.7. Does ID Vaccination Elicit Adverse Events More Frequently than IM or SC Route?

One of the main concerns regarding ID vaccination is that it may elicit adverse events more frequently than other administration routes. Of the included studies, 12 studies reported both local and systemic adverse events of influenza vaccines at the same dosage (15 µg of hemagglutinin antigen per strain) delivered either by ID, IM, or SC route [11,12,14,15,17,18,19,21,23,24,25,26]. Common local adverse events reported in these studies include pain, induration, erythema, swelling, pruritus, and ecchymosis. In general, ID vaccination elicited local adverse events more frequently than SC and IM vaccination with the ratios of any local adverse events of 1.61 and 2, respectively (Figure 5A,B). Pruritus, swelling, erythema, and induration were two times more frequent by the ID route than by SC route (Figure 5A), while the frequencies of these events were >3 times higher with the ID route compared to the IM route (Figure 5B). Meanwhile, pain and ecchymosis were observed at lower frequencies by ID route than SC or IM routes (Figure 5A,B).

Systemic adverse events, including fever, headache, malaise, myalgia, and arthralgia, were also reported at low frequency in these studies. These events were reported at mild conditions and there was no significant difference in the frequency of these events between ID and SC routes, and ID and IM routes (Figure 5C,D). There were no severe adverse events reported in these studies.

## 4. Discussion

Due to immunosenescence [3,4], the effectiveness of influenza vaccines is lower in elderly compared to younger groups [2]. To compensate for the reduced influenza-specific immune responses in this age group, MF59-adjuvant [5,6] and high-dose [7] influenza vaccines are recommended and achieve a certain level of increased immunogenicity. Changing the route of vaccine administration from conventional IM or SC routes to the ID route is another approach that has been investigated to enhance the immunogenicity of influenza vaccine in the elderly [8,9]. In theory, ID vaccination delivers vaccine antigens to the dermis, which contains an abundant supply of dermal dendritic cells and Langerhans cells, lymphatic vessels, accelerating or enhancing the transport of antigens to target T and B lymphocytes within the local lymph nodes [33,34,35]. However, in practice, ID influenza vaccine is not currently available for the elderly [1]. Therefore, the benefits of ID vaccination over conventional routes in the elderly need to be systematically evaluated. In this review, we compared the immunogenicity and the safety profile of ID and IM or SC immunization of influenza vaccines reported in the literature.

Overall, at the standard dose of 15 µg of hemagglutinin antigen per strain, ID vaccination was more immunogenic than IM and SC vaccinations in the elderly, confirming their higher immunogenicity predicted theoretically. ID influenza vaccines were also as immunogenic as MF59-adjuvanted IM influenza vaccine, implying that the immunogenicity of influenza vaccines could be enhanced just by changing the administration route of the vaccine without the need to change the vaccine formulation.

Interestingly, as shown in Figure 2B, at the standard dose of 15 µg of hemagglutinin antigen per strain, ID influenza vaccines were more immunogenic than IM vaccine of 45 µg of hemagglutinin antigen per strain [21], but less immunogenic than a high dose of 60 µg of hemagglutinin antigen per strain [24]. On the one hand, these results confirm that the immunogenicity of IM influenza vaccine is dependent on the concentration of hemagglutinin antigen. On the other hand, it may be argued that the immunogenicity of ID influenza vaccine could be further enhanced simply by increasing the concentration of hemagglutinin antigen formulated in ID influenza vaccines, as observed in IM influenza vaccines. However, higher concentrations of hemagglutinin antigen (21 µg) showed a minimal improvement in the immunogenicity of ID influenza vaccine (Figure 2A) [18,24]. In contrast, reducing the concentration of hemagglutinin antigen in ID influenza vaccine did not result in non-inferior influenza-specific immune responses as compared to standard-dose ID influenza vaccine [12,21]. A concentration of hemagglutinin antigen as low as 3 µg could be formulated in an ID influenza vaccine without a significant loss of immunogenicity [20]. These results together imply that the main benefit of ID administration of influenza vaccine in the elderly may lie in its capacity of eliciting non-inferior immune responses at reduced concentrations of hemagglutinin antigen rather than enhancing the immunogenicity. It appears that the activation threshold at the dermis is lower than that of other routes. It remains unclear whether the hemagglutinin concentration could be further reduced without sacrificing the immunogenicity of ID influenza vaccine and it would be interesting to address this question in future research. In addition, to the best of our knowledge, the effect of vaccine adjuvants has not been evaluated in ID influenza vaccines. This is a major knowledge gap that needs to be addressed. With the well-known effects of vaccine adjuvant such as M59 in IM influenza vaccines, it would be interesting to explore the effect of vaccine adjuvant formulated with a low amount of hemagglutinin antigen in ID influenza vaccine. Careful dosing may be needed to balance any changes in immunogenicity with potential increases in side effects or adverse events, already an elevated concern with the ID route.

The prime-boost regimen is a simple and common strategy to improve the immune responses in numerous vaccines. However, this strategy has limited benefit for influenza vaccines, with data from two studies showing a non-significant enhancement in the immunogenicity of two-dose vaccination as compared to one-dose vaccination [12,13]. Potential mechanisms underlying this failure remain unclear and thus, it is worth further investigation. Note that the influenza vaccines used for the prime-boost regimen in these studies were homologous. It would also be interesting to explore the potential effects of heterologous prime-boost regimen for influenza vaccination in the elderly.

## 5. Conclusions

In summary, our analyses showed that ID influenza vaccines were more immunogenic than IM and SC influenza vaccines at the same antigen dosage and as immunogenic as MF59-adjuvanted IM vaccines in the elderly. However, ID vaccines were less immunogenic than high-dose IM vaccines. Although ID administration of influenza vaccines elicited local adverse events more frequently than IM and SC routes, these events were under mild conditions. Our analyses also suggested that a major benefit of ID immunization of influenza vaccines mainly lies in its dose-sparing effect. Future research is still needed to optimize a more immunogenic ID influenza vaccine formulated from a low amount of hemagglutinin antigen and vaccine adjuvant for the elderly.

## Figures and Tables

**Figure 1 viruses-14-02438-f001:**
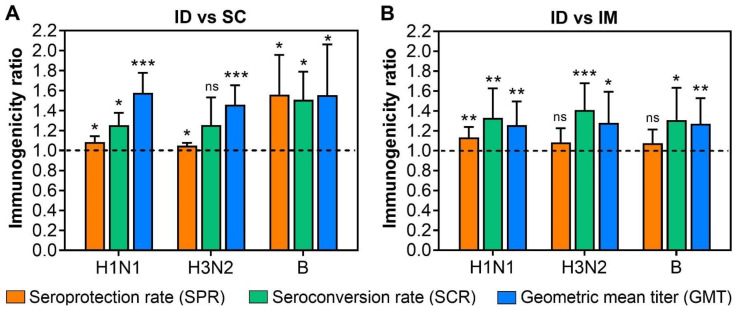
Comparison of the immunogenicity of standard-dose influenza vaccines delivered by intradermal (ID) and intramuscular (IM), or subcutaneous (SC) routes in the elderly. The ratios of seroprotection rate (SPR), seroconversion rate (SCR), and geometric mean titer (GMT) were calculated by dividing the mean SPR, SCR, and GMT of ID vaccination by those induced by IM or SC vaccinations. A ratio of >1 (dash line) implies greater immune responses induced by ID vaccination than IM or SC vaccinations. (**A**) At the same antigen dosage (15 µg of HA for each antigen), ID vaccination induced greater SPR, SCR, and GMT for each viral vaccine strain than SC vaccination. (**B**) Similarly, the SPR, SCR, and GMT elicited by ID vaccination were greater than those induced by IM vaccination. One sample *t*-test was used to examine the significant difference in the immunogenicity ratios against the hypothesized ratio of 1. “ns” = non-significant, “*” = *p* < 0.05, “**” = *p* < 0.01, “***” = *p* < 0.001.

**Figure 2 viruses-14-02438-f002:**
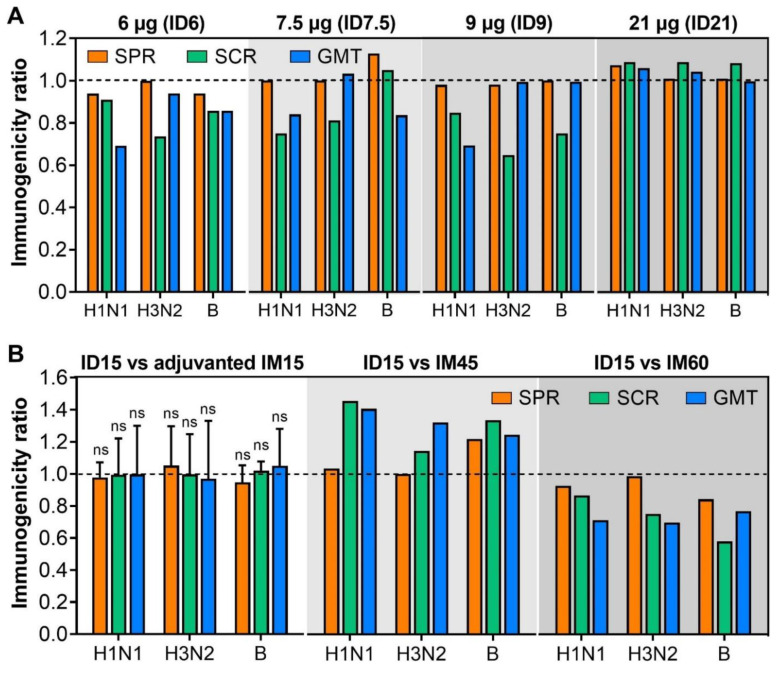
Comparison of the immunogenicity induced by varying amounts of hemagglutinin antigen in ID vaccines (**A**) and by standard-dose ID vaccines and MF59-adjuvanted IM15 and high-dose IM vaccines (B) in the elderly. (**A**) Comparison of immunogenicity induced by various dosages, including 6 µg (ID6), 7.5 µg (ID7.5), 9 µg (ID7.5), 21 µg (ID21) and standard dosage (15 µg) of influenza vaccines injected intradermally. The ratios of SPR, SCR, and GMT were calculated by dividing the mean SPR, SCR, and GMT of ID vaccination of various dosages by those of standard-dose ID15. A ratio of <1 (dash line) implies a lower immunogenicity of varying dosages of ID vaccination than ID15. (**B**) Comparison of immunogenicity of standard-dose ID15 and MF59-adjuvanted IM15, high dose (45 µg or IM45, 60 µg or IM60) of IM vaccines. The ratios of SPR, SCR, and GMT were calculated by dividing the mean SPR, SCR, and GMT of ID15 by those of adjuvanted IM15, IM45, and IM60. A ratio of >1 (dash line) implies a higher immunogenicity of ID15 over adjuvanted ID15, IM45, and IM60. One sample *t*-test was used to examine the significant difference in the immunogenicity ratios against hypothesized ratio of 1. “ns” = non-significant. Due to limited studies comparing the immunogenicity of ID15 and ID vaccines at varying amounts of hemagglutinin antigen, ID15 and high-dose IM vaccines, a statistical test was not performed.

**Figure 3 viruses-14-02438-f003:**
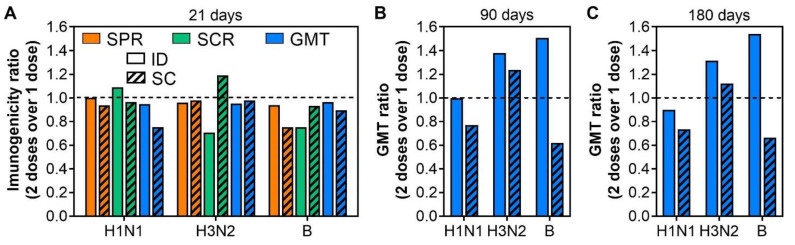
Comparison of immunogenicity induced by one-dose and two-dose regimen of influenza vaccine in the elderly. (**A**) Comparison of immunogenicity induced by one and two doses of ID15 and SC15 at 21 days after the last immunization. The ratios of SPR, SCR, and GMT were calculated by dividing the mean SPR, SCR, and GMT after immunization with two doses by those after one dose. A ratio of <1 (dash line) implies lower immunogenicity of two-dose regimen than one-dose regimen. Similar calculations for GMT ratio were performed at 90 days (**B**) and 180 days (**C**) after the last immunization. Due to limited studies comparing the immunogenicity of one-dose versus two-dose regimen, a statistical test was not performed.

**Figure 4 viruses-14-02438-f004:**
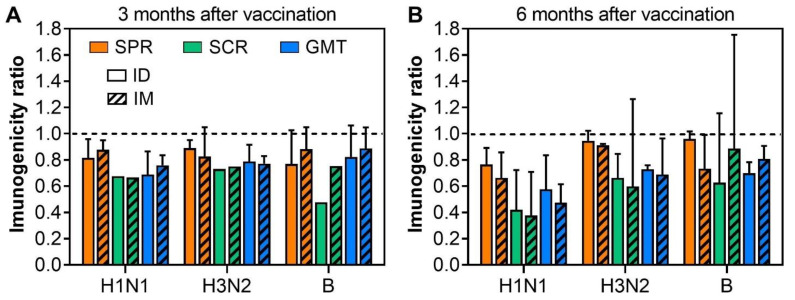
Comparison of the duration of influenza-specific immune responses at 3 (**A**) and 6 (**B**) months after vaccination in the elderly. The ratios of SPR, SCR, and GMT were calculated by dividing the mean SPR, SCR, GMT at 3 or 6 months by those at their peak of 21–28 days after the last immunization. A ratio of <1 (dash line) implies the waning of influenza-specific immune responses.

**Figure 5 viruses-14-02438-f005:**
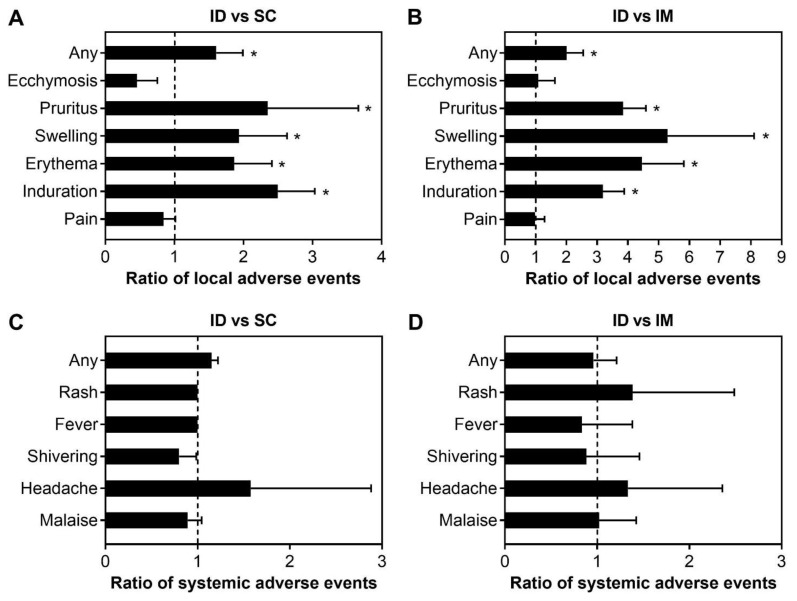
Comparison of the frequency of local and systemic adverse events elicited by ID versus SC vaccinations (**A**,**C**) and ID versus IM vaccinations (**B**,**D**) in the elderly. The ratios of the adverse events were calculated by dividing the frequency of these events elicited by ID route over SC or IM routes. A ratio of >1 (dash line) implies higher frequencies of local adverse events elicited by ID route. One sample *t*-test was used to examine the significant difference in the immunogenicity ratios against hypothesized ratio of 1 with “*” = *p* < 0.05. If not indicated, the difference in the ratio of adverse events was not statistically significant.

**Table 1 viruses-14-02438-t001:** Demographic characteristics of the study subjects. In the “Vaccine group” column, study subjects were vaccinated via either intradermal (ID), subcutaneous (SC), or intramuscular (IM) routes, with varying amounts of hemagglutinin, including 4.5, 6, 9, 15, 21, 45, and 60 µg of hemagglutinin for each vaccine viral strain (indicated as numbers after ID, SC, or IM). In the “Age” column, age is expressed either by median (range) or mean ± standard deviation (SD). In two studies [21,22], age and age distribution were not specified, although all study subjects in these two studies were ≥65 years. The “Time” column indicates the time of post-vaccination to assess the immune responses induced by influenza vaccine.

No.	Routes	Vaccine Group	No. of Subjects	Age	Time (Day)	Ref.
1	ID vs. SC	ID6	50	70.2 (65–80)	21	[12]
		ID9	50	70.5 (65–79)		
		ID15 (1 dose)	50	70.4 (65–79)		
		ID15 (2 doses)	50	69.9 (65–81)		
		SC15 (1 dose)	50	69.9 (65–79)		
		SC15 (2 doses)	50	69.9 (65–81)		
2	ID vs. SC	ID15	450	70 (65–88)	7, 21	[11]
		SC15	450	70 (65–82)		
3	ID vs. SC	ID15 (1 dose)	50	69 (67.3–73)	90, 180	[13]
		ID15 (2 doses)	50	70 (67–72.8)		
		SC15 (1 dose)	50	70 (66.3–72)		
		SC15 (2 doses)	50	69.5 (66–72)		
4	ID vs. IM	ID15	250	76.9 ± 8.4	28, 90	[14]
		IM15	250	75.3 ± 7.7		
5	ID vs. IM	ID15	110	66.1 ± 5.1	1, 30	[16]
		IM15	111	65.5 ± 3.8		
6	ID vs. IM	ID15	60	64.9 ± 3.6	21	[19]
		IM15	60	64.5 ± 3.8		
7		ID15	50	81.7 ± 7.8	21, 180	[17]
		IM15	50	83.9 ± 6.9		
8	ID vs. IM	ID15	359	70 (60–84)	21	[18]
		ID21	359	70 (60–85)		
		IM15	358	71 (60–85)		
9	ID vs. IM	ID15	398	73.9 ± 6.3	21	[26]
		MF59-adjuvanted IM15	397	74.7 ± 6.6		
10	ID vs. IM	ID15	636	73.1 ± 6.0	28, 180	[24]
		ID21	627	72.9 ± 5.9		
		IM15	317	73.4 ± 5.9		
		IM60	317	73.0 ± 6.0		
11	ID vs. IM	ID9	63	73.6 ± 6.3	28	[25]
		ID4.5 (2 doses)	65	74.7 ± 6.3		
		IM9	64	75.2 ± 7.7		
		IM15	65	75.6 ± 6.8		
12	ID vs. IM	ID7.5	61	≥65	22, 90, 180	[21]
		ID15 (Intanza)	60	≥65		
		ID15 (Inflexal)	61	≥65		
		IM15 (Inflexal)	63	≥65		
		MF59-adjuvanted IM15	63	≥65		
		IM45 (Inflexal)	62	≥65		
13	ID vs. IM	ID3	63	72 (68–77)	21	[20]
		ID9	68	73.5 (69.3–78)		
		IM15	66	72 (66–78)		
14	ID vs. IM	ID15	111	72 (65–86)	30, 180	[23]
		IM15	113	73 (65–88)		
		MF59-adjuvanted IM15	111	71 (65–88)		
15	ID vs. IM	ID15	303	73.7	21, 180	[22]
		IM15	307	73.8		
		MF59-adjuvanted IM15	301	73.9		
16	ID vs. IM	ID15	2606	71 (60–94)	21	[15]
		IM15	1089	71 (60–95)		
17	ID vs. IM	ID15	58	68.7 ± 5.5	21–28	[8]
		IM15	50	69.8 ± 7.6		

## Data Availability

All data synthesized from the literature and analyzed for the current study are available from the corresponding author upon reasonable request.

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
