# Peer review of "Enhancing Immunogenicity of Influenza Vaccine in the Elderly through Intradermal Vaccination: A Literature Analysis"

_viruses, 2022, doi:10.3390/v14112438_

Round 1

Reviewer 1 Report

The systematic review submitted by Huy Quang Quach and Richard B. Kennedy under the title "Enhancing immunogenicity of influenza vaccine in the elderly 2 through intradermal vaccination: a literature analysis" is written well and addressed the important issue of low immunogenicity of the currently available Influenza vaccines. I have a few suggestions to improve this review 

It will be good if authors add more studies in analysis. 

Discussion part first paragraph is almost the same as the introduction so improves the discussion. 

In conclusion, as authors studied many articles so they must be clear in the suggestion which route of administration, adjuvant or vaccine is best among the available. 

Author Response

Dear Reviewer,

Thank you very much for your time reviewing our manuscript and your constructive comments and suggestions. We strongly believe that your comments and suggestions greatly strengthen this manuscript. We have considered your comments/suggestions in depth and responded to each of your comments/suggestions in detail as below. We have also revised the manuscript accordingly and uploaded the revised manuscript for your reference. All changes in the revised manuscript were tracked. We have also uploaded our responses to the comments/suggestions from other reviewers for your information. Please do not hesitate to contact me if any questions arise.

Sincerely,

Richard B. Kennedy, Ph.D.

Reviewer 2 Report

In the abstract, semicolons were inappropriately used.  

In the abstract, in line 19, does the standard-dose influenza vaccine refers to the unadjuvanted influenza vaccine?

In line 19,  is the ID immunization with standard-dose influenza vaccine also as protective as IM immunization with adjuvanted influenza vaccine? Could you compare the in vivo or in vitro influenza protectiveness of these two vaccination strategies in laboratory animals or in clinical trials?

In line 21, suggest saying... between vaccinations in ID, IM, or SC route. 

line 41, ... main strategies have been investigated. investigated reads better. 

In line 45, suggest deleting "of" before "vaccine administration route...".  

In lines 50 - 51, the title of reference 10 reads more clearly. Here, before reading the whole article, it is not easy to understand the difference between the influenza vaccine and the standard dose. Is the antigen amount in ID vaccination standard dose or not? 

In line 152 figure 1, seroconversion, not serocoversion. 

In lines 171 -172, according to the subsequent depiction, it should be "as compared to a standard dose of 15 μg hemagglutinin antigen present in IM influenza vaccine".

In figure 2 legend, should it be adjuvanted IM15? 

Have the first and second strategies been applied in the marketed influenza vaccines for the elderly? Otherwise, in line 313, it is suggested to change "implemented" to "investigated". 

In lines 330 - 332, the article did not compare the immunogenicity of the MF59-adjuvanted vaccine that is delivered via ID route. Can the adjuvanted influenza vaccine by IM route be directly used for immunization in the ID route, Do you think so? 

A suggestion is to mention the elderly subjects in the first sentence of all figure legends. 

I am also interested in the distinct specific immunity strengths among the adjuvanted IM15, as well as the unadjuvanted IM45 and IM60 vaccination groups. As shown in figure 2B, the influenza vaccine immunogenicity shows the weakest in the IM45 group, while the strongest in the IM60 group. Could you estimate the amount of every vaccine antigen such as H1N1, H3N2, and Flu B, in IM vaccinations that would confer similar levels of specific immune responses as the ID15 group?  This can evaluate the efficacy of the ID immunization route in increasing vaccine immunogenicity in the elderly. 

In line 342, could you roughly define the spectrum of the lowered concentration?

In line 348, its capacity of eliciting non-inferior... 

In the last paragraph in the Discussion section, the boost immunization in one or two months did not enhance the specific immune responses. I am also curious about the ID vaccination efficacy in the next year, it is suggested to immunize the influenza vaccine every year. 

Is it massively reported on the vaccine efficacy, dosage, and specific immunity duration of intradermal influenza vaccination in aged laboratory animals? Could you please discuss these aspects and compare the ID and IM vaccinations in the aged laboratory animals? 

Please check the article and appropriately add "specific" prior to immune response or immunity. 

In line 374, suggest adding "vaccination" following "ID influenza". 

Author Response

(The authors gave the same response as above.)

Reviewer 3 Report

Influenza virus has been regarding as one of the potential pathogens for the next pandemics. Highly immunogenic flu vaccine is recommended to people, especially for the elderly to fight against the incoming dominant influenza virus every year. In this study, Quach and colleagues analyzed numerous studies published in the past 20 years in order to find out the best strategy to improve the immunogenicity of flu vaccine. With this in-depth literature analysis, they have proposed an ideal dose and route of administration to have the best immunogenic vaccine for the elderly. The study is appreciated, but number of major concerns must be addressed.

Major comments:

1.       Some sentences describe the significance of the findings, but proper statistical analysis is missing in the figures: Fig2, Fig 4 and Fig 5C and 5D. Therefore, the statement lacks support.

2.       Definition of the specific terms (eg. SPR, SCR, GMT) is recommended to be clearly stated. It helps reader to follow. SPR, SCR and GMT were selected for the comparison. What is the rationale for this selection? Why are these three factors selected for the comparison, but not two of them? What is the relationship between these three factors?

3.       In Fig 1, the immunogenicity of SCR varied across different vaccine strains. What is the implication? Further discussion is required.

4.       In section 3.1, the relationship between the immunogenicity of ID vaccine and the amount of hemagglutinin (HA) antigen was discussed and the minimal amount of HA antigen was proposed. The analysis did show that relationship, but the minimal amount of HA was proposed to be 3 µg, which was not based on Fig 2 or analysis in this review. If 3 µg is the ideal dose, corresponding analysis is required in this review. What is the difference between 3 µg and 6 µg, in terms of the SPR, SCR, and GMT ratio? Why is the GMT value not correlated to SPR value, given that GMT represents the vaccine titer while SPR represents the rate of antibody production?

5.       In Fig 2B, the fairness of the comparison of ID15 vs IM45, and ID15 vs IM60 is questionable because two variables have been changed in each comparison.

6.       Fig 3B, 3C and 4b are missing figure legend.

7.       Line255-259: the half-life of the antibody targeting H3N2 and influenza B is longer than that targeting H1N1 when the vaccine is administrated via ID. What is the implication? Why? Further discussion is required.

8.       When the vaccine is administrated via ID, more adverse effect is found. The elderly usually suffers from long-term diseases and complications. The adverse effect induced by ID vaccine would become a major concern. Are there any studies about the safety of ID vaccine to the elderly with long-term diseases and complications? Line 373: What are the criteria of the mild conditions to the elderly?

9.       In discussion section, some statements lack support by references or figures, eg. Line 334-336, 348-349.

 Minor comments:

1.       There is some typo- or grammatical error, eg. Line34, 44, 46, 47, 127,

2.       Line141: a new paragraph is recommended to discuss the variation of SPR ratio across different strains.

3.       Line 343: the sentence is quite complicated.

Author Response

(The authors gave the same response as above.)
